# Relationship Between Resilience and Fertility Quality of Life in Infertile Women: Mediating Roles of Infertility Self-Efficacy and Infertility Coping

**DOI:** 10.3390/healthcare13202589

**Published:** 2025-10-14

**Authors:** Jing Xu, Xin-Yuan Zhang, Yi-Bei Zhouchen, Ying Luo, Shi-Yun Wang, Sharon R. Redding, Yan-Qiong Ouyang, Dou Fu

**Affiliations:** 1School of Nursing, Wuhan University, Wuhan 430071, China; 2019203050026@whu.edu.cn (J.X.); 2023283070022@whu.edu.cn (X.-Y.Z.); zcyb1228@whu.edu.cn (Y.-B.Z.); 2023283070015@whu.edu.cn (Y.L.); 2022203070004@whu.edu.cn (S.-Y.W.); 2Project HOPE, Washington, DC 20006, USA; shredding1@gmail.com; 3Department of General Practice, Renmin Hospital of Wuhan University, Wuhan 430060, China

**Keywords:** infertility, resilience, self-efficacy, infertility coping, fertility quality of life

## Abstract

**Objectives**: This study aimed to explore the relationship between resilience, infertility self-efficacy, infertility coping strategies, and fertility quality of life (fertility QoL) in women with infertility. Additionally, it sought to examine the mediating roles of self-efficacy and infertility coping strategies in the relationship between resilience and fertility QoL. **Methods**: A cross-sectional study was conducted on infertile women undergoing assisted reproductive treatment from December 2021 to March 2022 in reproductive centers in Wuhan, China. A total of 314 participants were recruited using convenience sampling to complete a socio-demographic characteristics questionnaire, the 10-item Connor–Davidson Resilience Scale (CD-RISC-10), the Infertility Self-efficacy Scale (ISE), the Copenhagen Multi-Centre Psychosocial Infertility (COMPI) Coping Strategy Scale, and the Fertility QoL Scale. **Results**: Resilience (r = 0.375, *p* < 0.01), infertility self-efficacy (r = 0.584, *p* < 0.01), and meaning-based coping strategy (r = 0.191, *p* < 0.01) were positively correlated with fertility QoL. The other three infertility coping strategies, including active-avoidance coping (r = −0.367, *p* < 0.01), active-confronting coping (r = −0.143, *p* < 0.05), and passive-avoidance coping (r = −0.130, *p* < 0.05), were negatively correlated. The indirect effect of resilience on fertility QoL through three mediators, including infertility self-efficacy (*β* = 0.467, *p* < 0.001), active-confronting coping (*β* = −0.214, *p* < 0.001), and meaning-based coping (*β* = 0.161, *p* < 0.001), was significant (value of total indirect effect = 0.263, 95% CI, 0.188 to 0.350) with an effect of 71.5%. **Conclusions**: Resilience may be a psychological resource that promotes adaptive coping strategies. This allows women to be more confident in their management of infertility and treatment-related concerns, which promotes a better fertility QoL.

## 1. Introduction

Infertility is defined as the inability to conceive after 12 months or more of normal sexual activity without the use of any contraception [1]. The Centers for Disease Control and Prevention (CDC) of the United States emphasizes that infertility not only affects quality of life (QoL) but also brings about serious public health consequences, involving psychological distress, social stigma, financial strain, and marital breakdown [2]. According to the latest report from the World Health Organization (WHO) in 2023, approximately 17.5% of the global population will experience infertility at some point in their lives. This means that one in every six is affected, with the highest infertility rate observed in the Asia-Pacific region, at approximately 23.2% [3]. A large cross-sectional study involving 12,815 married women in China showed that the prevalence of infertility among Chinese childbearing married women reached 18% [4].

In every society, there is a value-norm system, and its members are expected to act in accordance with this system. In many societies, especially in China, childlessness can be seen as a departure from the norm or a stigma in society [5]. Infertility is a distressing experience that can disrupt individual developmental trajectories, marriages, and social relationships, leading to mood disturbances, anxiety, and depression [6]. Infertile women use a variety of strategies to cope with these problems, but some of the maladaptive coping strategies can even accelerate the deterioration of these problems [7,8]. For example, avoidance coping strategies can reduce fertility QoL [7].

Fertility Quality of Life (fertility QoL) is the perception and satisfaction of infertile individuals in all aspects of life in the context of infertility and fertility treatment settings [9,10]. Although both male and female factors can cause infertility, most women suffer from severe social pressure, environmental ridicule, and criticism [11,12]. In addition, most infertility treatments involve women, and the prevalence of mood disorders is higher in females than in males undergoing fertility treatment [13]. These factors cause deterioration of the QoL of infertile women and affect their treatment compliance and pregnancy rate [14].

Previous studies have mostly focused on the effects of negative psychology (stigma, stress) on the QoL of infertile women [9,15] (Jing et al., 2020; G. Li et al., 2020). In recent years, with the development of positive psychology, researchers have been given new perspectives and directions. Positive psychology emphasizes that psychology should focus on the positive aspects of individual positive experiences, including positive personal traits, well-being, and optimal functioning [16]. Resilience and self-efficacy, as positive psychological resources, have been extensively utilized in clinical settings and are known to positively influence the QoL of infertile couples [17,18,19,20].

A positive psychological perspective for individuals undergoing infertility treatment should focus on enhancing their strengths, well-being, and overall functioning [17,18]. Previous studies have demonstrated that coping strategies and self-efficacy mediate the relationship between resilience and QoL [21,22], but their specific influence on infertile women has not been well examined.

### 1.1. Purpose of This Study

From the perspective of positive psychology, this study explores the relationships between resilience, infertility self-efficacy, infertility coping strategies, and fertility QoL in women experiencing infertility. The following hypotheses were tested:

**Hypothesis** **1.**
*Resilience is positively related to fertility QoL, such that individuals with higher levels of resilience are more likely to have greater fertility QoL.*


**Hypothesis** **2.**
*Resilience is positively related to adaptive infertility coping strategies (such as seeking social support and active problem-solving) and negatively related to maladaptive strategies (such as avoidance and denial).*


**Hypothesis** **3.**
*Infertility self-efficacy and infertility coping strategies act as mediators in the relationship between resilience and fertility QoL, such that women with higher resilience are more likely to experience better fertility QoL due to enhanced self-efficacy and the use of more adaptive coping strategies.*


### 1.2. Theoretical Framework

This study is underpinned by the Broaden-and-Build Theory of Positive Emotions [23], which suggests that positive emotional experiences, even during times of stress, can enhance psychological resilience and contribute to improved coping strategies. Resilient individuals cultivate a broad array of psychological resources that help buffer negative emotions, thereby enabling more effective problem-solving and coping in stressful situations [24,25]. This resilience can be particularly important for women facing infertility, a life stressor that demands adaptive coping mechanisms. Based on the Broaden-and-Build Theory, we hypothesize that resilience may positively influence fertility QoL by enhancing infertility self-efficacy and promoting the adoption of more effective infertility coping strategies. The model suggests that resilient women are more likely to develop higher levels of self-efficacy and employ adaptive coping strategies when faced with infertility-related stress. These mediating factors, in turn, contribute to better fertility QoL.

## 2. Methodology

### 2.1. Study Design, Setting, and Samples

This was a cross-sectional study. From December 2021 to March 20, 2022, 314 participants were recruited from the infertility clinic/reproductive centers at two large hospitals in Wuhan, a city located in central China. The minimum sample size was 227, as calculated by *N* = (Z^2^ · *P* · (1 − *P*)/d^2^, based on Z = 1.96, *P* = 18% [26], 1 − *P* = 0.82, and d = 0.05. Considering a 15% loss rate, the final sample size should be a minimum of 268. The inclusion criteria were (a) women diagnosed as infertile and (b) age 20 through 45 years. The exclusion criteria were participants with mental disorders or cognitive impairment. This study was approved by the Ethics Committee of Wuhan University (2020YF0084).

### 2.2. Study Tools

#### 2.2.1. Socio-Demographic Characteristics Questionnaire

The researcher-designed questionnaire included age, ethnicity, employment status, education level, average monthly income per family member, number of children, the type of infertility, the duration of infertility diagnosis, the duration of fertility treatments, the form of fertility treatments, and the embryo transfer cycle.

#### 2.2.2. Connor–Davidson Resilience Scale (CD-RISC-10)

The 10-item CD-RISC is a brief self-rated instrument to help quantify resilience [27,28]. Each item is rated on a four-point scale, where 0 represents “not true at all” and 4 stands for “true nearly all the time.” The total score, ranging from 0 to 40, reflects an individual’s resilience, with higher scores indicating greater resilience. In this study, the Cronbach’s alpha for the scale was 0.895.

#### 2.2.3. Chinese Version of the Infertility Self-Efficacy Scale (ISE)

The Infertility Self-Efficacy Scale was designed to assess individuals’ confidence in their ability to manage infertility-related challenges and their attitudes toward treatment and diagnosis [29]. It consists of 16 items, each rated on a 9-point scale ranging from 1 (not at all confident) to 9 (completely confident). The total score spans from 16 to 144, with higher scores reflecting greater self-efficacy. The scale demonstrated excellent reliability, with a Cronbach’s alpha of 0.953 in this study.

#### 2.2.4. Copenhagen Multi-Centre Psychosocial Infertility (COMPI) Coping Strategy Scale

The COMPI was developed to measure infertility-related coping [30] and was categorized into four subscales, including active-avoidance coping, active-confronting coping, passive-avoidance coping, and meaning-based coping. In this study, Cronbach’s alphas of the four subscales were 0.716, 0.736, 0.725, and 0.754, respectively.

#### 2.2.5. Fertility Quality of Life Scale (FertiQoL)

This scale was designed to evaluate the fertility QoL of individuals experiencing fertility problems [31]. The scale includes 34 items, which are divided into two modules, including core FertiQoL and treatment FertiQoL. In addition, two items are used for the overall assessment of physical health and satisfaction with QoL and are not included in the FertiQoL score. The 34 items are scored using five response categories, ranging from 0 to 4, and three total scores (total FertiQoL score, core FertiQoL score, and treatment FertiQoL score), with a range of 0 to 100. Higher scores indicate greater fertility QoL. In this study, Cronbach’s alpha for the scale was 0.915.

### 2.3. Data Collection

The distribution of the questionnaires was completed by two trained team members, using uniform questionnaire instructions. Prior to completing the questionnaires, the individuals were informed of the purpose and significance of this study. After obtaining informed consent, a paper version of the questionnaire was distributed, and the participants were asked to complete it at that time. The participants received a small gift in thanks after completing the questionnaires.

### 2.4. Statistical Analysis

Statistical analyses were completed using SPSS 26.0 (IBM Corp., Armonk, NY, USA). Socio-demographic characteristics were described using the mean, standard deviation (SD), frequency, and percentage. Independent samples *t*-test and one-way analysis of variance (ANOVA) were conducted to evaluate differences in socio-demographic characteristics and treatment variables of the participants’ resilience, infertility self-efficacy, infertility coping strategy, and fertility QoL, respectively. Pearson correlation was used to analyze the associations among resilience, infertility self-efficacy, infertility coping strategy, and fertility QoL. Pearson correlation was used to analyze the associations among resilience, infertility self-efficacy, infertility coping strategy, and fertility QoL. The SPSS PROCESS V3.3 macro (model 4) developed by Hayes was conducted to explore whether infertility self-efficacy and infertility coping were the mediators in the relationship between resilience and fertility QoL [32]. The bootstrapping method was set to 5000 samples and was used to test whether direct and indirect effects were significant. The effect was significant if the 95% confidence interval (CI) did not contain zero. Statistical significance levels were set at less than 0.05 (two-tailed).

## 3. Results

### 3.1. Characteristics of the Participants

A total of 314 infertile women participated in this study. The average age of the participants was 32.23 (SD = 4.12) years. The socio-demographic characteristics of the participants are shown in Table 1.

### 3.2. Univariate Analysis of Resilience, Infertility Self-Efficacy, Infertility Coping Strategy, and Fertility Quality of Life

#### 3.2.1. Resilience

The mean score of resilience was 27.25 (SD = 6.14). The results showed that the scores of resilience differed regarding employment (*t* = −2.174, *p* = 0.030), education level (*F* = 4.965, *p* = 0.002), duration of infertility diagnosis (*F* = 3.039, *p* = 0.016), and duration of fertility treatments (*F* = 2.714, *p* = 0.037).

#### 3.2.2. Infertility Self-Efficacy

The mean score of infertility self-efficacy was 107.93 (SD = 22.14). The results indicated that the ISE score was significantly different regarding employment (*t* = −2.108, *p* = 0.036).

#### 3.2.3. Infertility Coping Strategy

The mean score for active-avoidance coping was 8.24 (SD = 2.33), with significant differences observed based on ethnicity (*t* = −2.032, *p* = 0.043) and monthly income (*F* = 2.533, *p* = 0.040). The mean score for active-confronting coping was 15.31 (SD = 3.96), showing no significant differences across the socio-demographic variables. The mean score for passive-avoidance coping was 8.37 (SD = 2.38), with significant differences found related to employment status (*t* = 2.106, *p* = 0.036) and monthly income (*F* = 3.326, *p* = 0.011). Finally, the mean score for meaning-based coping was 14.01 (SD = 3.29), with significant differences observed based on monthly income (*F* = 3.167, *p* = 0.014).

#### 3.2.4. Fertility Quality of Life

The mean total fertility QoL score was 68.52 (SD = 12.51). Significant differences were observed in the total fertility QoL score based on the duration of infertility diagnosis (*F* = 2.680, *p* = 0.037), the duration of fertility treatments (*F* = 14.768, *p* = 0.001), and the embryo transfer cycle (*F* = 4.421, *p* = 0.002). The core fertility QoL score, with a mean of 70.70 (SD = 13.92), showed significant variation concerning employment status (*t* = −2.147, *p* = 0.033), the duration of infertility diagnosis (*F* = 2.067, *p* = 0.048), the duration of fertility treatments (*F* = 14.059, *p* = 0.001), and the embryo transfer cycle (*F* = 4.695, *p* = 0.001). In terms of treatment fertility QoL, the mean score was 63.29 (SD = 12.08), with significant differences related to the duration of fertility treatments (*F* = 6.305, *p* = 0.002) and the embryo transfer cycle (*F* = 2.589, *p* = 0.037). The details are shown in Table 2.

### 3.3. Correlation Between Resilience, Infertility Self-Efficacy, Infertility Coping Strategy, and Fertility Quality of Life

Resilience, infertility self-efficacy, meaning-based coping, and fertility QoL were positively correlated with each other. The other three dimensions of infertility coping strategies (active-avoidance coping, active-confronting coping, and passive-avoidance coping) were not correlated with all the variables. The details are shown in Table 3.

### 3.4. Mediating Effect of Infertility Self-Efficacy and Infertility Coping Strategy on the Relationship Between Resilience and Fertility QoL

The socio-demographic variables of employment, duration of infertility diagnosis, duration of fertility treatments, and embryo transfer cycle were potential covariates in the mediation model (*p* < 0.05; see Table 2). The total effect of resilience on fertility QoL was significant (*β* = 0.368, *p* < 0.001; 95% CI, 0.272 to 0.476). The direct effect of resilience on fertility QoL was also significant (*β* = 0.105, *p* < 0.05; 95% CI, 0.272 to 0.476). Table 4 and Table 5 and Figure 1 indicate that the indirect effect of resilience on fertility QoL through three mediators, including infertility self-efficacy (*β* = 0.467, *p* < 0.001), active-confronting coping (*β* = −0.214, *p* < 0.001), and meaning-based coping (*β* = 0.161, *p* < 0.001), was significant (value of indirect effect = 0.263, 95% CI, 0.188 to 0.350), with an effect of 71.5%.

## 4. Discussion

The average score on total fertility QoL was (68.52 ± 12.15), which is similar to previous research on fertility QoL in infertile women in China, such as in Liaoning Province (64.54 ± 16.90) and Beijing (64.5 ± 14.1) [13,33]. In this study, the core fertility QoL (70.70 ± 13.92) was weaker than the treatment fertility QoL (63.29 ± 12.08) in the participants undergoing assisted reproductive therapy. Although assisted reproductive therapy is the most effective way to treat infertility, it also has serious negative effects on infertile couples. Assisted reproductive treatment has a long treatment cycle, is costly, includes invasive procedures, and the success rate is uncertain [34,35]. Moreover, repeated treatment failures lead women to lose confidence in treatment and themselves, ultimately creating negative psychological emotions and diminishing their treatment QoL [14].

The fertility QoL score was negatively correlated with the duration of infertility diagnosis, the duration of fertility treatments, and the embryo transfer cycle, which is consistent with the findings of previous studies [33,36]. Long-term treatment represents more treatment failures and setbacks, leading to greater self-doubt and psychological distress in women, which seriously threatens their QoL; an increase in the number of ART failures may lead to reduced happiness in infertile women and increased psychological and economic burdens [33]. Several studies have demonstrated that prolonged involuntary childlessness and expensive assisted reproduction have adverse effects on an individual’s fertility QoL [36,37].

The findings of the current study were consistent with those of several research studies, which found that resilience has direct and indirect effects on fertility QoL [21,38]. That is, infertile women have the ability to recover from an infertility diagnosis and treatment experience, which may help them reduce stress and improve their fertility QoL. Infertility is a devastating event for couples trying to conceive, with severe physical and psychosocial consequences for women, including depression, anxiety, stigma, stress, and loss of self-esteem [9,35,37]. When faced with infertility and fertility treatment, some resilient women can effectively manage and resolve the stress of infertility and achieve better reproductive outcomes [13,39]. Low-resilience women are overwhelmed by infertility events that impair their physical and mental functioning [39,40]. Commonly, for women with infertility, infertility treatment is a low-control situation. Efforts focused solely on finding solutions can lead to frustration, increased stress, and affect emotional instability. In contrast, emotion-focused coping strategies, which emphasize emotional regulation, acceptance, and emotional expression, are more adaptive in the low-control context of infertility treatment [13]. Women with higher resilience are more likely to effectively employ these strategies, enabling them to better cope with the psychological distress of infertility and maintain a higher quality of reproductive life [13]. Therefore, resilience is a determinant in improving the fertility QoL of infertile women [41].

Previous research has shown that self-efficacy plays a mediating role in the relationship between resilience and QoL among caregivers of stroke patients [21,22]. It enables individuals to better understand their condition and treatment, fosters confidence, enhances treatment adherence, and promotes healthy habits, all of which contribute to improved treatment outcomes and fertility QoL [41]. Pregnancy and treatment failures can undermine women’s confidence in managing infertility treatment, leading to self-doubt and reduced self-efficacy [42]. Those with low self-efficacy tend to focus on their limitations, making them more susceptible to negative emotions, anxiety, and depression [43]. In contrast, individuals with high self-efficacy positively manage their emotions, mitigating the adverse effects of negative emotions on fertility QoL [42,44].

Another study among women with gynecological cancers demonstrated that coping strategies mediate the relationship between resilience and QoL [21,22]. This study found that active-confronting coping strategies impair fertility QoL in infertile women, which is inconsistent with most studies. The study results of Jing et al. [9] were consistent with the findings in the current study, suggesting that an active-confronting coping strategy was negatively associated with fertility QoL in women undergoing infertility treatment. Adaptation or maladaptation of coping strategies employed by individuals depends on the type of health outcome and the characteristics of the stressor [45]. In the process of assisted reproductive treatment, women have to undergo various invasive procedures, endure complications of treatment, and bear a heavy financial burden and uncertainty about the success of pregnancy, which can lead to increased psychological distress and impaired fertility QoL [46]. Due to the uncontrollability of assisted reproductive therapy, women who adopt active-confronting coping strategies are more likely to experience treatment disappointment, which seriously affects their fertility QoL. Conversely, meaning-based coping strategies play a positive mediating role in the relationship between resilience and fertility QoL, which is consistent with previous studies in which meaning-based coping strategies were beneficial for better fertility quality and lower stigma [4]. Women who used meaning-based coping strategies chose to redefine their life goals and values, which had a significantly positive effect on the marital benefits of infertile couples and attenuated the negative effects of infertility on fertility QoL in infertile women [38]. Previous research has also shown that meaningful coping can improve the marital relationship and fertility QoL of infertile women [10,47].

This study further found that infertility coping strategies and infertility self-efficacy jointly mediate the relationship between resilience and fertility QoL. Individuals with high resilience will stimulate high self-efficacy and tend to choose positive and meaningful coping strategies to deal with infertility-related stress. A high level of infertility self-efficacy and the use of a meaning-based coping strategy can prevent infertile women from experiencing more negative emotions, thereby improving fertility QoL, while an active-confronting coping strategy can undermine the fertility QoL. Therefore, health care providers should integrate resilience into clinical practice interventions to improve fertility QoL in infertile women. When resilience is difficult to improve, healthcare providers can compensate for the effect of resilience on fertility QoL by encouraging infertile women to respond meaningfully to infertility events or by increasing their self-efficacy.

## 5. Strengths and Limitations

This study examined the impact of positive psychological resources (resilience, efficacy, and positive coping strategies) on fertility QoL in infertile women from a positive psychology perspective. Moreover, this study confirmed that infertility self-efficacy and infertility coping strategies have a mediating effect on the relationship between resilience and fertility QoL in women with infertility.

However, there are several limitations that should be acknowledged. First, while this study explored the effects of resilience, infertility self-efficacy, and infertility coping strategies on the quality of life of infertile women, factors such as family structure (e.g., nuclear vs. joint families) and living environment (e.g., rural vs. urban areas) were not addressed. Additionally, the details of etiologies, hormonal levels, and the body mass index (BMI) were not considered. Future research should further investigate these factors. Second, the data were collected through self-reported questionnaires, which are subject to potential biases based on the participants’ mental state at the time of data collection. To improve the reliability and validity of future studies, it would be beneficial to involve professional psychological assessments or consultations with psychologists. Finally, as infertility and treatment are issues that couples need to face together, they have a strong correlation with each other in terms of infertility-related distress and psychological issues. This study only explored the influence of infertile women’s own psychological factors on their quality of life, ignoring their partners’ factors. Future research should explore the impact of infertile couples’ positive psychological resources on their own and their partners’ fertility QoL.

## 6. Implications

The current study provides some empirical evidence for future clinical research and intervention on fertility QoL in women with infertility. It is critical to implement interventions regarding fertility QoL in infertile women. Infertility and infertility treatment cause physical pain and psychological stress in women with infertility and seriously endanger their fertility QoL, which affects their treatment compliance and success rates. In addition, the findings of the current study demonstrated that resilience was an effective predictor of fertility QoL. Interventions targeting the fertility QoL of infertile women should incorporate an improvement in resilience. Moreover, for low-resilience women with infertility, fertility QoL can be improved from the perspective of increasing meaningful infertility coping and infertility self-efficacy.

## 7. Conclusions

This study demonstrates that assisted reproductive treatment affects the fertility QoL of women with infertility. Resilience, infertility self-efficacy, and meaning-based coping were positively correlated with fertility QoL. Conversely, the other three infertility coping strategies (active-avoidance coping, active-confronting coping, and passive-avoidance coping) were negatively correlated with fertility QoL. Resilience may act as an individual inner psychological resource, allowing infertile women to conduct adaptive infertility coping strategies and be more confident in their ability to deal with infertility and treatment-related problems, enabling them to attain a higher fertility QoL. Future research should focus on the impact of infertile women’s inherent positive psychological resources on fertility QoL and integrate these resources to improve fertility QoL.

## Figures and Tables

**Figure 1 healthcare-13-02589-f001:**
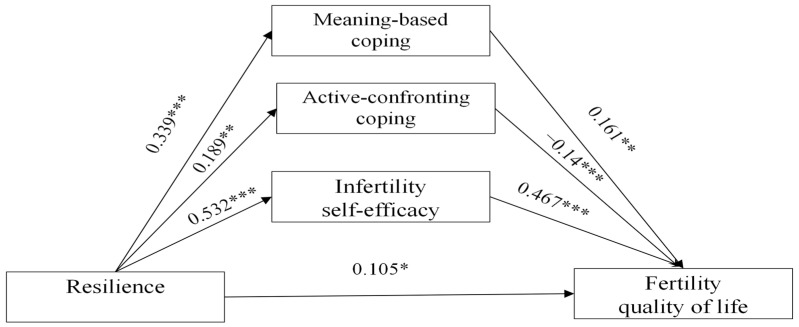
Model of the mediating effect of infertility self-efficacy, active-confronting coping, and meaning-based coping in the relationship between resilience and fertility quality of life (standardization). All pathways were statistically significant * *p* < 0.05; ** *p* < 0.01; *** *p <* 0.001.

**Table 1 healthcare-13-02589-t001:** Socio-demographic characteristics of the participants (*N* = 314).

Variables	Category	Frequency	Constituent Ratio/Rate (%)
Age (years)	≤35	259	82.5
	>36	55	17.5
Ethnicity	Han nationality	307	97.8
	Minority	7	2.2
Employment	Employed	260	82.8
	Unemployed	54	17.2
Education Level	Junior high school or below	39	12.4
	High school	64	20.4
	College or university	160	51.0
	Master’s degree or higher	44	14.0
	No data	7	2.2
Average monthly income per family member (CNY)	<5000	63	20.1
	5000 to 10,000	148	47.1
	10,001 to 15,000	54	17.2
	>15,000	35	11.1
	No data	14	4.5
Number of children	0	259	82.6
	1	46	14.6
	2	8	2.5
	No data	1	0.3
Type of infertility	Primary	179	57.0
	Secondary	133	42.4
	No data	2	0.6
Duration of infertility diagnosis (years)	≤1	156	49.7
	1 to 2	53	16.9
	2 to 3	55	17.5
	>3	49	15.6
	No data	1	0.3
Duration of fertility treatments (years)	≤1	208	66.2
	1 to 2	39	12.4
	2 to 3	30	9.6
	>3	33	10.5
	No data	4	1.3
Embryo transfer cycle	0	197	62.7
	1	76	24.2
	2	23	7.3
	3	10	3.2
	>3	8	2.6

Abbreviations: IUI: intrauterine insemination; IVF-ET: in vitro fertilization–embryo transfer.

**Table 2 healthcare-13-02589-t002:** Univariate analysis of resilience, infertility self-efficacy, infertility coping strategy, and fertility QoL of the participants (*N* = 314).

Variables	Resilience (Mean ± SD)	ISE (Mean ± SD)	COMPI (Mean ± SD)				FertiQoL (Mean ± SD)		
			Active-Avoidance	Active-Confronting	Passive-Avoidance	Meaning-Based	Total FertiQoL	Core FertiQoL	Treatment FteriQoL
Total	27.25 ± 6.14	107.93 ± 22.14	8.24 ± 2.33	15.31 ± 3.96	8.37 ± 2.38	14.01 ± 3.29	68.52 ± 12.51	70.70 ± 13.92	63.29 ± 12.08
Age (years)									
≤35	27.28 ± 6.12	108.4 ± 21.76	8.21 ± 2.35	15.21 ± 3.85	8.44 ± 2.34	14.08 ± 3.26	68.71 ± 12.02	70.93 ± 13.64	63.38 ± 12.19
>36	27.11 ± 6.28	105.46 ± 23.93	8.39 ± 2.26	15.80 ± 4.44	8.06 ± 2.57	13.68 ± 3.46	67.61 ± 12.82	69.60 ± 15.29	62.82 ± 11.62
*t*	0.190	0.911	−0.525	−1.002	1.074	0.815	0.608	0.640	0.312
*p*	0.849	0.363	0.600	0.317	0.284	0.416	0.544	0.523	0.756
Ethnicity									
Han nationality	27.20 ± 6.08	107.93 ± 22.13	8.20 ± 2.32	15.33 ± 3.98	8.34 ± 2.38	13.97 ± 3.29	68.49 ± 12.19	70.70 ± 13.94	63.20 ± 12.12
Minority	29.57 ± 8.79	108.00 ± 24.38	10.00 ± 2.31	14.54 ± 3.22	9.71 ± 2.06	15.57 ± 3.31	69.85 ± 11.24	70.98 ± 13.89	67.14 ± 9.83
*t*	−1.012	−0.008	−2.032	0.518	−1.513	−1.272	−0.293	−0.054	−0.853
*p*	0.313	0.993	0.043 *	0.605	0.131	0.204	0.770	0.957	0.394
Average monthly income per family member (CNY)									
No data	26.86 ± 7.17	118.92 ± 21.47	7.00 ± 2.32	13.36 ± 3.69	6.57 ± 2.62	11.46 ± 4.49	69.95 ± 10.95	69.79 ± 13.67	70.36 ± 8.76
<5000	25.71 ± 6.69	101.56 ± 24.78	8.76 ± 2.30	14.99 ± 3.53	8.97 ± 2.53	13.65 ± 3.23	66.07 ± 12.93	67.97 ± 15.40	61.51 ± 12.23
5000 to 10,000	27.13 ± 5.86	108.50 ± 21.23	8.35 ± 2.30	15.34 ± 3.89	8.32 ± 2.18	14.09 ± 3.03	69.29 ± 11.91	71.66 ± 13.62	63.62 ± 11.49
10,001 to 15,000	28.26 ± 6.24	109.51 ± 21.45	7.76 ± 1.91	16.01 ± 4.20	8.13 ± 2.34	14.80 ± 3.30	67.74 ± 11.78	70.51 ± 12.67	61.11 ± 12.83
>15,000	29.11 ± 5.18	110.16 ± 20.36	8.06 ± 2.86	15.49 ± 4.58	8.60 ± 2.56	14.11 ± 3.50	70.34 ± 12.67	72.29 ± 14.37	65.64 ± 13.16
*F*	2.218	2.389	2.533	1.399	3.326	3.167	1.089	0.910	2.381
*p*	0.067	0.051	0.040 *	0.234	0.011 *	0.014 *	0.362	0.458	0.052
Education level									
Junior high school or below	24.95 ± 6.46	103.59 ± 23.62	7.74 ± 2.19	14.92 ± 3.65	8.85 ± 2.46	13.56 ± 3.55	67.44 ± 12.56	68.40 ± 14.45	65.13 ± 10.97
High school	26.48 ± 5.98	106.19 ± 22.10	8.57 ± 2.45	15.33 ± 4.06	8.62 ± 2.47	13.78 ± 3.44	67.96 ± 11.46	69.45 ± 14.03	64.38 ± 11.91
College or university	27.32 ± 6.02	108.54 ± 22.45	8.28 ± 2.24	15.17 ± 3.63	8.33 ± 2.20	14.12 ± 3.10	68.91 ± 12.38	71.46 ± 14.02	62.80 ± 11.69
Master’s degree or higher	29.84 ± 5.67	111.23 ± 19.83	8.22 ± 2.54	16.48 ± 4.99	8.05 ± 2.74	14.59 ± 3.44	68.65 ± 12.10	71.52 ± 12.85	61.76 ± 14.43
*F*	4.965	0.990	1.025	1.465	1.007	0.847	0.205	0.739	0.797
*p*	0.002 *	0.398	0.382	0.224	0.390	0.469	0.893	0.529	0.496
Employment									
Employed	25.85 ± 6.02	102.59 ± 22.71	8.59 ± 2.24	15.90 ± 3.36	8.95 ± 2.28	13.90 ± 3.27	66.11 ± 12.65	67.23 ± 15.05	63.40 ± 11.98
Unemployed	27.72 ± 6.16	109.16 ± 22.05	8.16 ± 2.37	15.17 ± 4.16	8.26 ± 2.35	14.03 ± 3.34	68.99 ± 11.95	71.42 ± 13.48	63.15 ± 12.18
*t*	−2.174	−2.108	1.299	1.282	2.106	−0.290	−1.689	−2.147	0.142
*p*	0.030 *	0.036 *	0.195	0.201	0.036 *	0.772	0.092	0.033 *	0.887
Number of children									
0	27.31 ± 6.16	107.60 ± 22.44	8.24 ± 2.38	15.40 ± 4.01	8.37 ± 2.34	14.11 ± 3.27	68.03 ± 12.42	70.13 ± 14.17	62.99 ± 12.36
1	26.63 ± 5.80	109.63 ± 20.36	8.26 ± 2.08	14.89 ± 3.46	8.59 ± 2.40	13.65 ± 3.34	70.88 ± 11.03	73.44 ± 12.60	64.73 ± 10.71
2	29.75 ± 7.38	112.00 ± 23.63	8.43 ± 2.64	15.57 ± 5.44	7.43 ± 3.55	13.21 ± 4.00	70.69 ± 8.91	72.02 ± 12.30	67.50 ± 7.64
*F*	0.906	0.295	0.024	0.335	0.736	0.590	1.183	1.135	0.829
*p*	0.405	0.745	0.976	0.715	0.480	0.555	0.308	0.323	0.438
Type of infertility									
Primary	27.29 ± 6.10	109.60 ± 22.57	8.17 ± 2.22	15.39 ± 3.81	8.44 ± 2.28	14.12 ± 3.07	68.90 ± 12.06	71.34 ± 13.25	63.04 ± 13.02
Secondary	27.18 ± 6.19	105.89 ± 21.52	8.37 ± 2.46	15.29 ± 4.13	8.33 ± 2.50	13.91 ± 3.58	67.83 ± 12.25	69.56 ± 14.67	63.67 ± 10.65
*t*	0.153	1.464	−0.751	0.231	0.393	0.542	0.767	1.117	−0.455
*p*	0.878	0.144	0.453	0.817	0.695	0.588	0.444	0.265	0.650
Duration of infertility diagnosis (years)									
<1	26.78 ± 5.98	107.64 ± 22.67	8.50 ± 2.17	14.94 ± 3.55	8.02 ± 2.35	14.12 ± 3.19	68.03 ± 8.13	72.33 ± 12.25	63.84 ± 12.32
1 to 2	27.62 ± 7.05	110.68 ± 21.95	8.72 ± 2.50	15.30 ± 4.02	8.88 ± 2.19	14.39 ± 3.51	69.98 ± 9.10	72.62 ± 12.88	63.63 ± 12.08
>2	30.47 ± 5.52	109.28 ± 20.37	8.93 ± 2.34	15.60 ± 4.10	9.27 ± 4.08	14.56 ± 3.51	72.35 ± 8.42	70.28 ± 13.26	65.76 ± 14.87
*F*	3.039	1.521	2.608	2.221	1.429	0.311	2.680	2.067	1.462
*p*	0.016 *	0.220	0.075	0.110	0.271	0.281	0.037 *	0.048 *	0.087
Duration of fertility treatments (years)									
< 1	27.69 ± 6.03	107.28 ± 22.45	8.26 ± 2.53	16.23 ± 4.58	8.28 ± 2.52	14.12 ± 3.80	69.36 ± 11.53	72.02 ± 12.87	64.36 ± 11.86
1 to 2	25.69 ± 6.96	109.72 ± 19.30	8.45 ± 1.99	16.25 ± 4.33	8.54 ± 2.59	14.22 ± 3.82	68.48 ± 11.15	70.51 ± 12.36	60.90 ± 12.97
>2	27.93 ± 4.03	110.05 ± 21.22	8.57 ± 2.57	16.92 ± 4.56	8.77 ± 2.85	14.38 ± 3.77	64.73 ± 11.33	66.65 ± 13.98	63.01 ± 12.06
*F*	2.714	1.625	1.357	2.400	2.673	0.378	14.768	14.059	6.305
*p*	0.037 *	0.197	0.258	0.091	0.069	0.687	0.001 *	0.001 *	0.002 *
Embryo transfer cycle									
0	27.65 ± 6.11	108.91 ± 22.04	8.03 ± 2.36	15.04 ± 3.87	8.06 ± 2.33	13.79 ± 3.25	70.16 ± 10.78	72.80 ± 12.19	63.84 ± 11.46
1	26.58 ± 6.38	107.56 ± 22.71	8.53 ± 2.38	15.55 ± 4.21	8.86 ± 2.34	14.64 ± 3.23	67.25 ± 14.09	68.33 ± 16.21	64.67 ± 12.87
2	26.13 ± 5.99	102.17 ± 21.72	8.13 ± 2.14	14.87 ± 3.91	8.39 ± 2.73	13.00 ± 3.67	66.14 ± 12.01	68.75 ± 13.70	59.89 ± 13.74
3	28.90 ± 5.43	105.70 ± 18.20	9.40 ± 1.07	16.90 ± 3.11	9.50 ± 2.22	15.50 ± 2.17	61.32 ± 13.31	63.33 ± 16.07	56.50 ± 10.22
>3	24.88 ± 5.36	106.75 ± 27.30	9.50 ± 2.20	19.00 ± 3.02	9.88 ± 1.81	14.38 ± 4.14	56.25 ± 13.54	56.77 ± 16.16	55.00 ± 11.34
*F*	1.111	0.517	1.935	2.566	3.052	2.014	4.421	4.695	2.589
*p*	0.351	0.723	0.104	0.038	0.017	0.092	0.002 *	0.001 *	0.037 *

Abbreviations: ISE: infertility self-efficacy; COMPI: Copenhagen Multi-Centre Psychosocial Infertility; FertiQoL: fertility quality of life. * *p <* 0.05.

**Table 3 healthcare-13-02589-t003:** Inter-correlations among the main study variables.

Variables	1	2	3.1	3.2	3.3	3.4	4
1 Resilience	1						
2 ISE	0.509 **	1					
3.1 Active-avoidance	−0.038	−0.215 **	1				
3.2 Active-confronting	0.135 *	0.052	0.400 **	1			
3.3 Passive-avoidance	−0.013	−0.096	0.375 **	0.503 **	1		
3.4 Meaning-based	0.283 **	0.258 **	0.334 **	0.530 **	0.510 **	1	
4 FertiQoL	0.375 **	0.584 **	−0.367 **	−0.143 *	−0.130 *	0.191 **	1

Note: **, correlation is significant at the 0.01 level (2-tailed); *, correlation is significant at the 0.05 level (2-tailed).

**Table 4 healthcare-13-02589-t004:** Regression analysis of the series mediation model (standardization).

Variables	Model 1 FertiQoL		Model 2 ISE		Model 3 Active-Confronting Coping		Model 4 Meaning-Based Coping		Model 5 FertiQoL	
	*β*	*t*	*β*	*t*	*β*	*t*	*β*	*t*	*β*	*t*
Employment	0.095	0.755	0.1351	1.0865	−0.227	−1.599	−0.047	−0.342	−0.009	−0.088
Duration of infertility diagnosis	0.169	2.419 *	0.0763	1.0991	−0.027	−0.336	0.030	0.385	0.123	2.070 **
Duration of fertility treatments	−0.322	−4.121	−0.0366	−0.4723	0.141	1.599	0.050	0.586	−0.283	−4.246 ***
Embryo transfer cycle	−0.138	−2.351 ***	−0.0294	−0.504	0.122	1.834	0.058	0.896	−0.108	−2.147 **
Resilience	0.368	7.046 ***	0.532	10.256 ***	0.189	3.202 **	0.339	5.921 ***	0.105	1.981 *
ISE									0.467	9.079 ***
Active-confronting coping									−0.214	−4.137 ***
Meaning-based coping									0.161	2.985 ***
*R* ^2^	0.247		0.285		0.072		0.115		0.463	
*F*	18.826 ***		22.665 ***		4.432 ***		7.401 ***		30.375 ***	

Abbreviations: FertiQoL, fertility quality of life; ISE, infertility self-efficacy. * *p* < 0.05; ** *p* < 0.01; *** *p* < 0.001; two-tailed.

**Table 5 healthcare-13-02589-t005:** Bootstrap results of the mediating effect of ISE, active-confronting coping, and meaning-based coping in the relationship between resilience and fertility quality of life.

Type of Effect	Value	Boot SE	Boot 95%CI	
			LLCI	ULCI
Total effect	0.368	0.052	0.272	0.476
Direct effect	0.105	0.055	−0.002	0.215
Indirect effect				
Total	0.263	0.042	0.188	0.350
ISE	0.248	0.039	0.178	0.331
Active-confronting coping	−0.040	0.017	−0.078	−0.012
Meaning-based coping	0.055	0.023	0.014	0.104

Abbreviations: ISE: infertility self-efficacy.

## Data Availability

The datasets generated and analyzed during the current study are not publicly available because the data are also part of an ongoing study; however, the datasets are available from the corresponding author upon reasonable request.

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
