# Peer review of "Relationship Between Resilience and Fertility Quality of Life in Infertile Women: Mediating Roles of Infertility Self-Efficacy and Infertility Coping"

_healthcare, 2025, doi:10.3390/healthcare13202589_

Round 1

Reviewer 1 Report

Comments and Suggestions for Authors

Dear Author(s), 

Your manuscript is very interesting and important, however kindly see below some suggestions of improvement:

  • The Introduction section should better underline the literature gap and why this topic is important, especially for China, or the region itself.
  • It would also be highly recommended to have a main aim of the study and then some specific objectives. The introduction section should end with the aim and the objectives of the study as clear as possible. 
  • I feel the need of a theoretical background in which the major concepts are explained and the hypotheses of the mediation model are explained. Otherwise, you can take passages from the discussion section and build a theoretical background. I am referring to this because I do not understand how some concepts may be converted or act as mediators. 
  • in the methodology section, t-tests and ANOVA were used for the differences, but I would recommend authors to add a sentence in which they mention testing the quantitative variables with normality tests as for instance, Shapiro-Wilk in order to be able to use the t-tests, otherwise the nonparametric tests should be used. Please include this into the results section. 

Thank you and good luck!

Author Response

1.The Introduction section should better underline the literature gap and why this topic is important, especially for China, or the region itself.

Response: We thank the reviewer for this helpful suggestion. In response to the suggestion regarding the Introduction section, we have revised paragraphs 1, 2, and 4 to better highlight the literature gap and explain why this topic is important, particularly for China and the surrounding region. The modified text has been marked in red font in the revised manuscript.

2.It would also be highly recommended to have a main aim of the study and then some specific objectives. The introduction section should end with the aim and the objectives of the study as clear as possible.

Response: Thank you for your valuable suggestion. Your input has made the content of the article much clearer. We have already clearly outlined the specific objectives of the study in Section 1.1. "Purpose of the Study" (in red font).

3.I feel the need of a theoretical background in which the major concepts are explained and the hypotheses of the mediation model are explained. Otherwise, you can take passages from the discussion section and build a theoretical background. I am referring to this because I do not understand how some concepts may be converted or act as mediators.

Response: Thank you very much for your valuable suggestion. I completely agree with your point. We have already addressed the theoretical framework of the study in Section 1.2. "Theoretical Framework," where we explain how the variables in this study are transformed and act as mediators (in red font).

4.In the methodology section, t-tests and ANOVA were used for the differences, but I would recommend authors to add a sentence in which they mention testing the quantitative variables with normality tests as for instance, Shapiro-Wilk in order to be able to use the t-tests, otherwise the nonparametric tests should be used. Please include this into the results section.

Response: We thank the reviewer for this important comment. We completely agree with your point. We have already made normality tests prior to data ananlysis and we assure that the data was normally distributed.

Reviewer 2 Report

Comments and Suggestions for Authors

The authors analyzed the relationship between Resilience and Fertility Quality of Life in Infertile Women.

This is a well-written cross-sectional study that explores an important topic within reproductive psychology.

The study applies a positive psychology framework to understand the mechanisms through which resilience might improve the fertility quality of life in women undergoing infertility treatment. The findings are valuable and could inform clinical interventions. The study successfully addresses a gap , the need for more research from a positive psychology perspective to explore how positive psychological resources interact to influence outcomes.

Here are my comments:

  • Introducation is short , and lacks aim.
  • The sample is somehow heterogeneous, as for treatment : explain how coping to Traditional Chinese medicine therapy could be compared to suregery or IUI procedures ?
  • In results, there are too many subtitles and tables are long. CHoose to keep only tables and reduce writings or try to summarize several table's data in text.
  • Avoid redundance
  • No need to precise each item for which duration eg : Embry transfer cycles or duration of infertility
  • Explain more figure 1 in the title.
  • Table 2 is long and involves a large number of statistical tests, increasing the risk of false positives. Consider applying a correction for multiple comparisons.
  • in the discussion, add more details for aspects like , in a low-control situation like infertility treatment, problem-focused strategies can become ineffective and frustrating, whereas emotion focused strategies become more adaptive. 
  • Add in the limitations the lack of details on etiologies, hormonal details, BMI etc ..

Author Response

1.Introducation is short , and lacks aim.

Response: We sincerely appreciate your insightful feedback. We have expanded the Introduction section and clearly outlined the aim of the study in Section 1.1, "Purpose of the Study" (in red font).

2.The sample is somehow heterogeneous, as for treatment : explain how coping to Traditional Chinese medicine therapy could be compared to suregery or IUI procedures ?

Response: Thank you for your concern. The data is somewhat complex, as there is overlap between different treatment protocols. Nearly all IVF-ET patients undergo Ovulation Induction and Follicle Monitoring. After careful consideration by the research team, we believe that these sample characteristics do not significantly impact the meaning of the study, and we are considering removing the related content. I ask for your kind understanding in this matter.

3.In results, there are too many subtitles and tables are long. CHoose to keep only tables and reduce writings or try to summarize several table's data in text.

Response: Thank you very much for your thoughtful suggestion. In response to your feedback, I have kept the tables, removed the excessive subtitles, and provided a summary of the data within the text to improve clarity and conciseness.

4.No need to precise each item for which duration eg : Embry transfer cycles or duration of infertility.

Response: Thank you for your valuable suggestion. In response, we have streamlined the time categories for the "Duration of Infertility Diagnosis" and "Duration of Fertility Treatments" in Table 2, Univariate Analysis of Resilience, Infertility Self-Efficacy, Infertility Coping Strategy, and Fertility QoL of Participants (in red font).

5. Explain more figure 1 in the title.

Response: Thank you for your comments. In response, I have provided a more detailed explanation of Figure 1 in the title, as per your recommendation.

6. Table 2 is long and involves a large number of statistical tests, increasing the risk of false positives. Consider applying a correction for multiple comparisons.

Response: Thank you for your concern. We attempted to apply a correction for multiple comparisons, but after adjustment, we found that it did not achieve the desired effect. However, after reviewing Table 2, our research team has removed unnecessary data and retained only the most essential information. I ask for your kind understanding in this matter.

7. in the discussion, add more details for aspects like , in a low-control situation like infertility treatment, problem-focused strategies can become ineffective and frustrating, whereas emotion focused strategies become more adaptive.

Response: Thank you for your thoughtful suggestion. In response, We have added more details regarding the aspects, specifically in the third paragraph of the discussion section (in red font), where we elaborated on how, in a low-control situation like infertility treatment, problem-focused strategies can become ineffective and frustrating, while emotion-focused strategies become more adaptive.

8.Add in the limitations the lack of details on etiologies, hormonal details, BMI etc ..

Response: Thank you for your instructions. In response, we have added the lack of details of etiologies, hormon level, BMI, and other related factors as limitations  in Section 5, "Strengths and Limitations." (in red font).

Reviewer 3 Report

Comments and Suggestions for Authors
  1. In the abstract Line no. 24 to 26 sentence may be re-structured, the objective is not clear, the authors can break the sentence into two.
  2. The methodology section is not clearly explained. “G*Power 3.1.9.7, based on a medium effect size (f2) of 0.15, significance level 74 α of 0.05, power (1-β) of 0.95, and 23 predictors” this statement could have been self-explanatory for the readers to understand
  3. The inclusion and exclusion criteria are wage, speaking mandarin in a medical study may not be a vital inclusion criteria, same way in the exclusion criteria , donor can be automatically ignored since it is illegal but the patient is already depressed with occupation or family or societal pressure, or whether she is taking any anti-depression or already lost a baby and become infertile or more abortions lead to infertility. Since the authors are working on QoL, they should have these background information as a questionnaire or as patient history
  4. Line No. 73 says “The minimum sample size was 234 “ but in the abstract it says “A total of 314 participants were recruited” , Finally how many participants involved in the study?
  5. Line no. 80 can be removed “2.2. Instruments”, since this study doesn’t involve any instrument , you may remove it or u may use the term “tools”
  6. The questionnaire was given in the printed paper format or Google form type digital form?
  7. The authors can follow same abbreviation throughout the manuscript, fertility QoL, in some places they are using
  8. In the results section, too many tables with too many numbers and data, the authors could have convert few table into a graphical representation could have been more appealing for the readers to understand as well.
  9. The authors could have planned their study “rural vs urban” or “nuclear family vs joined family”, they play major role in the stress & depression.
  10. The total study is based only on questionnaire and the reliability of this study is totally depends on the volunteers mental state on that day of data collection, hence there may be false positive or negative, instead if a psychologist report or opinion or consultation involved could have given better reliable results
  11. Over all, it is a well-planned study and executed well. Addition of some clinical data could have been strengthen this study more.
Comments on the Quality of English Language

Some sentences are very long and complicated.  These sentences can be split into two or three statements. 

Author Response

1.In the abstract Line no. 24 to 26 sentence may be re-structured, the objective is not clear, the authors can break the sentence into two.

Response: Thank you for your helpful suggestion. In response, I have restructured the sentence in lines 24 to 26 of the abstract to improve clarity and ensure that the objective is stated more clearly (in red font).

2.The methodology section is not clearly explained. “G*Power 3.1.9.7, based on a medium effect size (f2) of 0.15, significance level 74 α of 0.05, power (1-β) of 0.95, and 23 predictors” this statement could have been self-explanatory for the readers to understand.

Response: Thank you for your valuable suggestion. Our research team discussed this issue. In response, we recalculated the sample size using a formula-based approach and describe it in Section 2, "Methodology," specifically in Section 2.1, "Study Design, Setting, and Samples." (in red font).

3.The inclusion and exclusion criteria are wage, speaking mandarin in a medical study may not be a vital inclusion criteria, same way in the exclusion criteria , donor can be automatically ignored since it is illegal but the patient is already depressed with occupation or family or societal pressure, or whether she is taking any anti-depression or already lost a baby and become infertile or more abortions lead to infertility. Since the authors are working on QoL, they should have these background information as a questionnaire or as patient history.

Response: Thank you for your insightful suggestion. In response, we have revised the inclusion and exclusion criteria as per your recommendation. The relevant content has been updated in Section 2, "Methodology," specifically in Section 2.1, "Study Design, Setting, and Samples." (in red font).

4.Line No. 73 says “The minimum sample size was 234 “ but in the abstract it says “A total of 314 participants were recruited” , Finally how many participants involved in the study?

Response: Thank you for your concern. Initially, based on the G*Power calculation, the minimum required sample size was 234. However, in the actual study, we recruited a total of 314 participants. Therefore, a total of 314 participants were involved in the study.

5.Line no. 80 can be removed “2.2. Instruments”, since this study doesn’t involve any instrument , you may remove it or u may use the term “tools”.

Response: Thank you very much. In response, we have updated the section title from "2.2. Study Instruments" to "Study Tools," as per your recommendation (in red font).

6.The questionnaire was given in the printed paper format or Google form type digital form?

Response: Thank you for your question. The questionnaire in this study was completed in paper format on-site after obtaining participants' informed consent. For further details, please refer to section in 2.3, "Data Collection."

7.The authors can follow same abbreviation throughout the manuscript, fertility QoL, in some places they are using.

Response: Thank you for your attention to detail. In response, we have standardized the abbreviation "fertility QoL" throughout the manuscript.

8.In the results section, too many tables with too many numbers and data, the authors could have convert few table into a graphical representation could have been more appealing for the readers to understand as well.

Response: Thank you for your concern. As you mentioned, there are indeed many tables in the results section. We have tried converting some of the tables into graphical representations, but the results did not meet our expectations. Therefore, we are humbly appealing to retain the tables and hope for your understanding.

9.The authors could have planned their study “rural vs urban” or “nuclear family vs joined family”, they play major role in the stress & depression.

Response: Thank you very much for your valuable suggestion. We will certainly consider incorporating factors such as "rural vs urban" or "nuclear family vs joint family" in future studies. We have also mentioned this in Section 5, "Strengths and Limitations." (in red font).

10.The total study is based only on questionnaire and the reliability of this study is totally depends on the volunteers mental state on that day of data collection, hence there may be false positive or negative, instead if a psychologist report or opinion or consultation involved could have given better reliable results.

Response: Thank you very much for your thoughtful suggestion. We have considered your point and, in response, have included it inin Section 5, "Strengths and Limitations." (in red font). We also agree that involving a psychologist's report or consultation could have given better reliable results.and we will consider this approach in future studies.

Round 2

Reviewer 1 Report

Comments and Suggestions for Authors

All suggestions of improvement have been addressed properly. 

Thank you!